# Development of Combined Heavy Rain Damage Prediction Models with Machine Learning

**Changhyun Choi [1], Jeonghwan Kim [2,\*], Jungwook Kim [1] and Hung Soo Kim [3]** 

[1] Institute of Water Resources System, Inha University, Michuhol-Gu, Incheon 22212, Korea; karesma0cch@naver.com (C.C.); rlawjddnr1023@gmail.com (J.K.)

[2] Department of Statistics, Ewha Womans University, Seodaemun-gu, Seoul 03760, Korea

[3] Department of Civil Engineering, Inha University, Michuhol-Gu, Incheon 22212, Korea; sookim@inha.ac.kr

\* Correspondence: sinkei9456@naver.com; Tel.: +82-32-874-0069

**Abstract:** Adequate forecasting and preparation for heavy rain can minimize life and property damage. Some studies have been conducted on the heavy rain damage prediction model (HDPM), however, most of their models are limited to the linear regression model that simply explains the linear relation between rainfall data and damage. This study develops the combined heavy rain damage prediction model (CHDPM) where the residual prediction model (RPM) is added to the HDPM. The predictive performance of the CHDPM is analyzed to be 4–14% higher than that of HDPM. Through this, we confirmed that the predictive performance of the model is improved by combining the RPM of the machine learning models to complement the linearity of the HDPM. The results of this study can be used as basic data beneficial for natural disaster management.

**Keywords:** disaster management; heavy rain damage; machine learning; natural disaster; prediction model; residual prediction model

## 1. Introduction

The intensity and frequency of extreme events has increased worldwide due to global warming [1–4]. In particular, rapid urbanization and industrialization have led to the increase in asset values as well as population growth in disaster-vulnerable areas thereby aggravating damages [5,6]. If we can identify the damage trends based on historical data and develop a model that can predict the magnitude and extent of damage, it can be of great help in devising measures to reduce harm [7,8]. More specifically, information on the predicted damages will serve as the basic data to support decision-making on disaster preparation and prevention. Studies that developed prediction models for natural disaster damages include linear regression models using independent variable like maximum wind speed, movement speed of typhoons, antecedent rainfall, total rainfall, and snowfall [9–14]. In addition, some studies pursued to improve predictive performance by including social and economic factors as independent variables such as area, population, income level, number of houses, and financial independence [15–20]. Research trials were also made using weather data as independent variables while grouping the social and economic characteristics of a study area to provide a damage prediction function [21,22]. As machine learning techniques (e.g., for image analysis and time series data processing [23–27]) have been proven to improve predictive performance, many researchers have also used it for predicting natural disasters [28–32]. However, most of the past studies have used a single model, but recently studies are showing more interest in combining models with different characteristics to improve the predictive performance [33–35].

Most of the previous studies used a linear regression model that draws and explains the linear relationship between weather phenomena and damage. However, such a simple expression of linear

relationship has a limitation not just in understanding the complex mechanisms of damage occurrence but also in accurately predicting the actual damage costs. Although different machine learning techniques have been used for developing prediction models to improve their performance, very few have used the combined models. Therefore, this study aims to develop a combined heavy rain damage prediction model by adding a residual model to the existing linear regression model using machine learning techniques.

This study used 30-year damage cost data due to heavy rains as the dependent variable for the heavy rain damage prediction model (HDPM). On the other hand, the rainfall data that directly affect the damage were used as independent variables for the linear regression model. The study also used residuals, which are the difference between the actual and predicted values derived from the linear regression model, as the dependent variables of the residual prediction model (RPM). The RPM was developed using machine learning models (decision tree, random forest, support vector machine, and deep neural networks) with socio-economic data (gross regional domestic product, financial independence rate, population density, population, area, dilapidated dwelling rate, number of houses, number of dilapidated dwellings, processing capacity of pumps, number of pumps) as independent variables. The sum of the predicted residuals from the RPMs and the predicted values from the HDPM were used for the final predicted values of the combined heavy rain damage prediction models (CHDPMs). The predictive performance of the HDPM and the CHDPMs were evaluated using the predictive performance indicators. Figure 1 depicts the flow chart of HDPM and CHDPM.

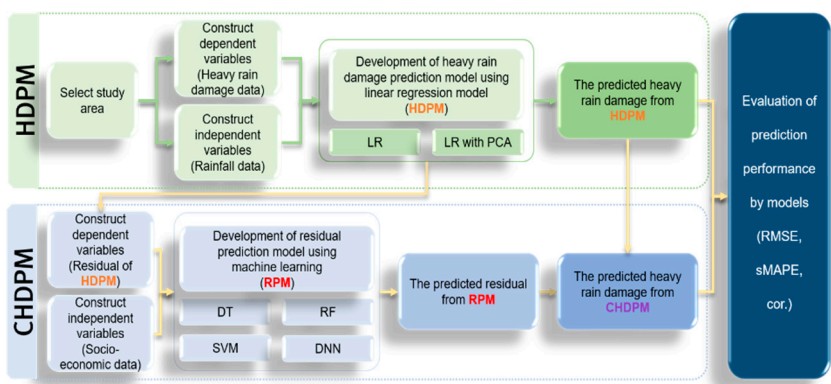

**Figure 1.** Flow chart. Abbreviations: HDPM: heavy rain damage prediction model; LR: linear regression; PCA: principle component analysis; RPM: residual prediction model; DT: decision tree; SVM: support vector machine; RF: random forest; DNN: deep neural network; CHDPM: combined heavy rain damage prediction model; RMSE: root mean square error; sMAPE: symmetric mean absolute percentage error.

## 2. Materials and Methods

### 2.1. Linear Regression Model

The linear regression (LR) model assumes the linear relation between the dependent variable $y$ and $k$ independent variables, $x_1, x_2, \cdots, x_k$.

$$y_i = \beta_0 + \beta_1 x_{1i} + \beta_2 x_{2i} + \cdots + \beta_k x_{ki} + \varepsilon_i \tag{1}$$

In the equation above the subscript $i$ means the $i$th data. $\beta_0$ is for intercept, while $\beta_1, \beta_2, \cdots, \beta_k$ are regression coefficients for the independent variables and $\varepsilon$ is the error that the model cannot explain. When independent variables are highly correlated in a linear regression model, it involves a problem of multicollinearity that distorts the estimation of individual regression coefficients. For the trained regression model, $k$ number of variance inflation factors (VIFs) can be obtained. VIFs are measures of the linear correlation between $x_j$ ($j$th variable) and $x_{(-j)}$ (remainder) for $j = 1, \ldots, k$.

VIF values above 10 indicate multicollinearity which is a strong linearity among independent variables. Multicollinearity can make the estimates of regression coefficients biased and the model's prediction unstable. The principal component regression model can be considered as a way of dealing with multicollinearity. The principal component regression model takes a small number of principal components that are independent of each other, instead of a large number of highly correlated independent variables. There are two ways used for determining the number of principal components: (1) In the scree plot, which presents the cumulative ratio for the total variance of a set of independent variables sequentially from the first principal component, select the number of principal components when the slope changes abruptly; or (2) select the number of principal components when the total cumulative dispersion ratio described meets a predetermined threshold. The study used the the latter and selected the number of principal components with 90% value as the threshold for the total accumulative variance.

*2.2. Machine Learning*

2.2.1. Decision Tree

Decision tree (DT) expresses data, with a graph in the shape of tree, based on rules and conditional statements of variables. According to a classification rule, similar data are sub-grouped and this process is iteratively repeated until the final criterion is satisfied [36]. In particular, a decision tree which intends to solve a regression problem is called a "regression tree". The decision tree finds a rule that can best explain dependent variables by partitioning the space of each independent variable repeatedly. Such binary recursive partitioning identifies a branching variable or point that minimizes the mean squared error (MSE) for each of the steps. The process is repeated to ultimately form a whole tree. In the case of a regression tree, the prediction is performed as follows: For a new data point of independent variables $x$, the trained tree finds the final node containing $x$ and presents the average of the response values in that node as the predicted value. To improve the predictive performance, pruning some branches of the tree can be considered. Pruning makes the grown tree not just simpler and easier to interpret but also helps to provide more accurate predictions. Figure 2 shows an example of a decision tree that completed the learning, a DT is more advantageous to use because it makes intuitive interpretation of a model possible. The study used the "rpart" library for the decision trees provided by the statistics software "R".

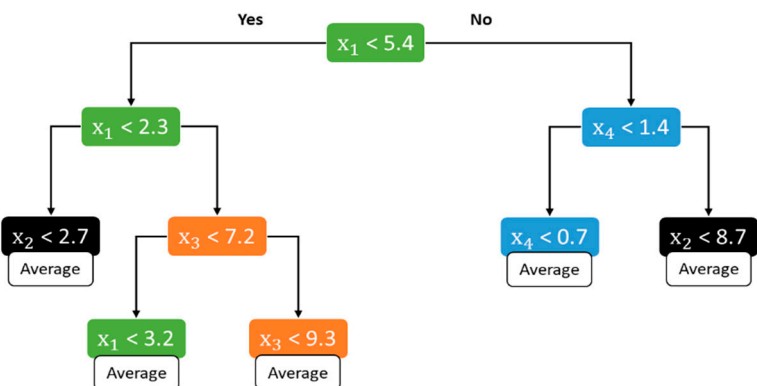

**Figure 2.** Example of a decision tree.

2.2.2. Random Forest

Random forest (RF) is one of the ensemble models that develops a number of prediction models and combines them to get the final prediction model [37]. The RF generates several bootstrap samples (training data) from the original data and trains decision trees in each bootstrap samples by using only some of the independent variables. Then when new data points are given for independent variables,

final predictions are provided after averaging or voting the predictions of the bootstrap trees. In the regression problem of RF, averaging is used. Figure 3 shows the prediction process of the RF with new data points given for independent variables.

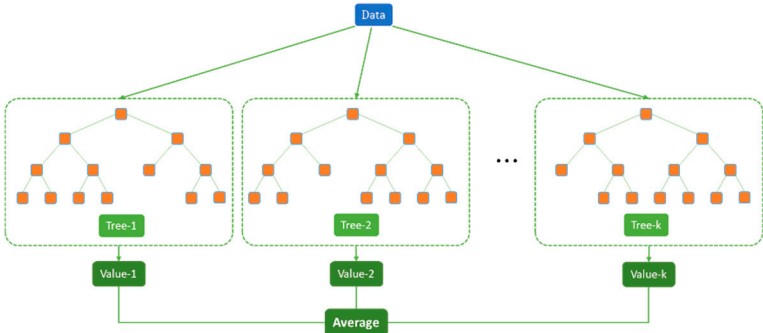

**Figure 3.** Example of random forest.

The OOB (out-of-bag) error can be used to optimize the number of bootstrap samples that consist the RF. The OOB error is a value that evaluates predictive performance of the bootstrap trees using the remaining data that are not included in the bootstrap samples. The model's predictive performance can be improved by using the optimal number of bootstrap samples where the OOB error is minimized. In the same way, the number of independent variables for each bootstrap tree can be optimized. When the purpose of prediction is regression, the default value for the number of independent variables is 1/3 of the total independent variables [37]. Unlike the decision tree, in which the interpretation is intuitive, the RF is a black box model for which interpretation is not possible. However, based on the contributions of independent variables in each bootstrap tree, the variable importance can be obtained; and accordingly the independent variables with relatively high importance can be identified. The study used the "randomForest" library for the RF provided by the statistical software "R".

### 2.2.3. Support Vector Machine

Support vector machine (SVM) is a machine learning model proposed by [38]. The SVM was originally intended to solve the problem of classification as a way to find optimal hyperplane composed of support vectors that can classify the vectors of different classes by the maximum margin for the distance between them. Figure 4 shows the separation of different classes with the maximum margin for the case where linear separation is applicable.

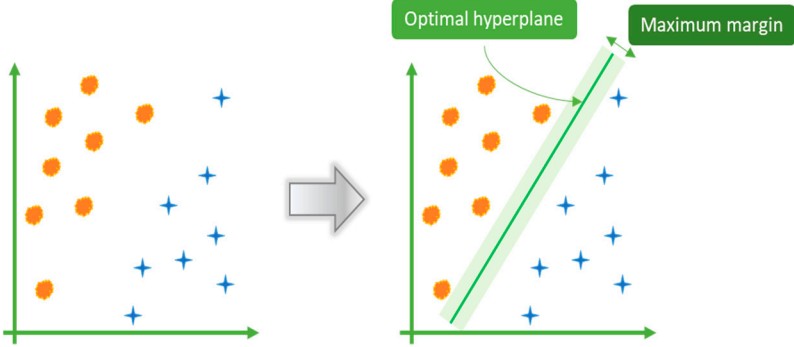

**Figure 4.** Example of support vector machine.

In the case of data for which the linear separation is not applicable, the kernel function was used to map them from the plane to a higher dimensional space before the classification. The polynomial, sigmoid, and radial basis function (RBF) are examples of kernel functions. Then, the SVM was expanded with the introduction of $\varepsilon$-insensitive loss function to be used for regression analysis and this

is called support vector regression (SVR). In this study the "e1071" library for the SVR was provided by the statistical software "R".

### 2.2.4. Deep Neural Networks

Artificial neural network (ANN) is a machine learning model inspired by a biological neural network. The ANN consists of one or more hidden layers, nodes, and weights. These components connect the input layer (for the independent variable) and the output layer (for the dependent variable). The ANN with three or more hidden layers is called a deep neural network (DNN). An example is shown in Figure 5.

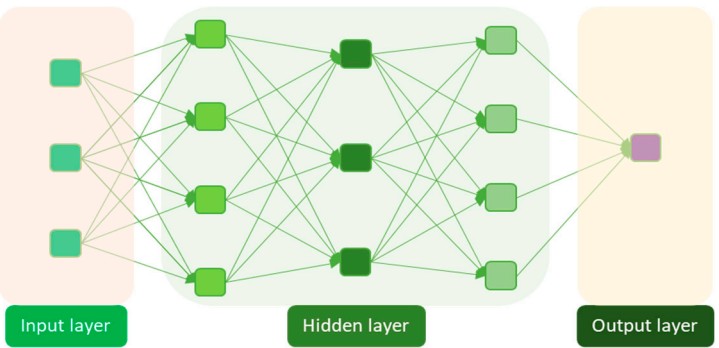

**Figure 5.** Example of deep neural networks.

The weights of the neural network model are trained through the back propagation algorithm. This is needed to produce a prediction close to the actual value of the dependent variable. The hidden layer node is a new feature created through the operation of several weights and nonlinear transformations from the input layer. It is known as an algorithm where the feature extraction is automatic. In addition, the consecutive non-linear conversion between hidden layers uses an algorithm that is advantageous in estimating the complex non-linear relationship between the independent variables and the dependent variables. Before the 1980s, the ANN was not widely used due to poor computer performance. However, recently, advances in computing technology make it possible to conduct an in-depth learning of the network. The active research and development (R&D) investment of large companies such as Google and Amazon also led to the rapid growth of the field under the name of "deep learning". The representative deep learning techniques include the deep neural network (DNN), convolutional neural network (CNN), and recurrent neural network (RNN). Most of the data used by the CNN and RNN are characterized by image data and time series data. However, the data for this study is neither images nor time series, thus, we considered the use of the DNN to be more appropriate. The study used the "neuralnet" library for the DNN provided by the statistical software "R".

### 2.3. Proposal of Combined Heavy Rain Damage Prediction Model

The dependent variable, $y$, is the heavy rain damage cost that is applied with the common logarithms ($\log_{10}$) to reduce data variability, while the independent variables, $X = (x_1, x_2, \cdots, x_k)$ are the rainfall data that has direct influence on heavy rain damages (maximum rainfall of duration from 1 h to 24 h, total rainfall, and antecedent rainfall from −1 day to −7 days). The study defined the heavy rain damage prediction model (HDPM) as the model that assumes the relation between the dependent variables and independent variables (Equation (1) where $\varepsilon$ is an error term).

$$\text{HDPM}: y_i = f_1(X_i) + \varepsilon_i, \ f_1(X_i) = \beta_0 + \beta_1 x_{1i} + \cdots + + \beta_{ki} x_{ki} \tag{2}$$

The $f_1(\cdot)$ of Equation (2) can be trained with the linear regression model discussed in Section 2.1. As reviewed in past studies, the HDPM assumes the linear relation between the dependent and

independent variables. They also showed a limitation in explaining the complex non-linear relation between heavy rain damages and rainfall data. Accordingly, significant gaps (residuals) tend to occur between the actual and predicted damage. Therefore, it was considered that the use of the prediction of the residuals of HDPM would greatly help to produce a result close to the actual.

This study used machine learning that secures a relatively high ability to explain a complex non-linear relation. The predicted residuals derived from the residual prediction model (RPM) were combined with the predicted damage costs provided by the existing HDPM. This aims to propose an advanced way for estimating damage prediction. To this end, the study considered the RPM that has the residual, $e_i = y_i - \hat{f}_1(X_i)$, obtained after the training the HDPM as the dependent variable and social and economic data as independent variables $Z = \left(z_1, z_2, \cdots, z_p\right)$ (in the equation $\delta$ indicates an error term).

$$\text{RPM} : e_i = f_2(Z_i) + \delta_i \tag{3}$$

The $f_2(\cdot)$ of Equation (3) can be trained with the machine learning model introduced in Section 2.2. The predicted residual, $\hat{f}_2(Z_i)$, calculated with the trained machine learning model was combined with the prediction of the HDPM as a final result. The study defined the algorithm that improves predictive performance of the linear regression model (HDPM) through the machine learning model (RPM) as the combined heavy rain damage prediction model (CHDPM). Steps for the CHDPM algorithm are as follows:

Step 1. Training of HDPM

From the original data $D_1 = \{X_i, y_i\}_{i=1}^n$ which contains the dependent variable $y$ and the independent variable $X$, training is made for the linear regression model below ($\varepsilon$ indicates an error term).

$$y_i = f_1(X_i) + \varepsilon_i \, , \, f_1(X_i) = \beta_0 + \beta_1 x_{1i} + \cdots + +\beta_k x_{ki}$$

The prediction of the HDPM for the new data $x$ which will be used for the independent variable $X$ is calculated as $\hat{f}_1(x)$.

Step 2. Training of RPM

From the total data $D_2 = \{Z_i, e_i\}_{i=1}^n, e_i = y_i - \hat{f}_1(X_i)$ which has the residual $e$, derived from the HDPM training as the dependent variable, and the independent variable $Z$ which is not used in Step 1, training is made for the machine learning model below ($\delta$ indicates an error term).

$$e_i = f_2(Z_i) + \delta_i$$

When new data for $X$ and $Z$ are given with $x$ and $z$, respectively, the prediction value of the CHDPM is calculated as the $\hat{f}_1(x) + \hat{f}_2(z)$ that combines the results of the HDPM and the RPM.

*2.4. Evaluation of Predictive Performance by Models*

To prevent overfitting of the developed heavy rain damage prediction model, the study randomly divided the total dataset into two groups: one for the training dataset and the other for test dataset. The increase of predictive performance of the test dataset raises the expectation for the excellent predictive performance of the models for future data. For this reason, the study developed the prediction model using the training dataset only, and then applied it to the test dataset to compare the actual with the predicted heavy rain damage for the evaluation of predictive performance. In this study, the root mean square error (RMSE), symmetric mean absolute percentage error (sMAPE), and correlation coefficient (cor.) were used as the predictive performance evaluation measures to compare the predicted and actual values. The equation for each of the measures are shown from Equation (4) to Equation (6). The RMSE means a standard deviation of residuals, which are the differences between predicted and actual values. It indicates how much error is included in the predicted, in comparison with the actual. Meanwhile, sMAPE is an indicator that shows the percentage of errors in predicted values. Therefore,

when the RMSE and the sMAPE approach zero (0), the model has a better predictive performance. The cor. is an indicator that shows the level of correlation between actual and predicted values with its range from −1 to 1. The cor. value closer to one (1) indicates a stronger positive correlation between the two datasets and has a better predictive performance.

$$\text{RMSE} = \sqrt{\frac{\sum_{i=1}^{m}(y_i - \hat{y}_i)^2}{m}} \tag{4}$$

$$\text{sMAPE} = \frac{1}{m}\sum_{i=1}^{m}\frac{|y_i - \hat{y}_i|}{(|y_i| + |\hat{y}_i|)/2} \tag{5}$$

$$\text{cor.} = \frac{\sum_{i=1}^{m}(y_i - \overline{y})(\hat{y}_i - \overline{\hat{y}})}{\sqrt{\sum_{i=1}^{m}(y_i - \overline{y})^2}\sqrt{\sum_{i=1}^{m}(\hat{y}_i - \overline{\hat{y}})^2}} \tag{6}$$

In the equations above, $y_i$ and $\hat{y}_i$ indicate the actual value and predicted value, respectively. Additionally, $\overline{y}$ and $\overline{\hat{y}}$ mean the average of the actual values and prediction values, respectively. The predictive performance evaluation process can be briefly explained below (see also Figure 6):

1. The total dataset is classified into a training dataset (70%) and test (30%) dataset.
2. A model is developed from the training dataset and is applied to the test dataset.
3. For each of the models (HDPM and CHDPM), a comparison is made for predictive performance using predictive performance evaluation measures.

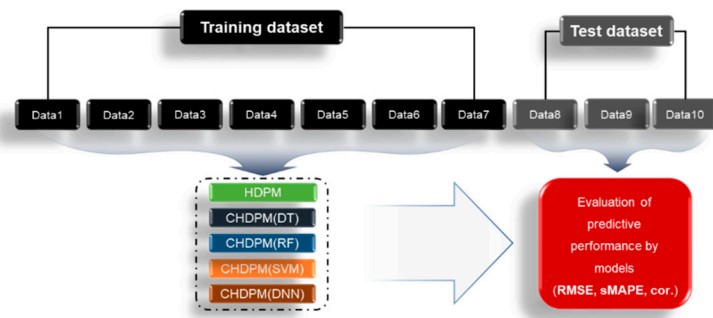

**Figure 6.** Concept of evaluation of prediction performance.

## 3. Heavy Rain Damage Prediction Model

### 3.1. Study Area

To select the study area, we referred to the "Statistical Yearbook of Natural Disaster" and surveyed the history and current tendency of heavy rain damages for 30 years (from 1988 to 2017). The Statistical Yearbook of Natural Disaster (published by the Ministry of the Interior and Safety, MOIS) provides statistical data on disasters. This yearbook combines records related to natural disasters and is normally issued at the end of every year, in accordance with the "Countermeasures against Natural Disasters Act." Table 1 summarizes the number of heavy rain damage events and heavy rain damage costs for each of the 17 *si* (cities) and *do* (provinces) in South Korea for 30 years. For the number of heavy rain damage events by *si* and *do*, Gyeonggi-do experienced the most frequent damages (in Bold) due to heavy rains, followed by Jeollanam-do and Gyeongsangbuk-do. In these three provinces, more than 500 cases of damage have been reported for the last 30 years. Gangwon-do showed the highest total damage cost, followed by Gyeonggi-do, Gyeongsangnam-do, and Chungcheongbuk-do. In these four provinces, the total damage cost for 30 years reached more than 1 trillion Korean Won (KRW).

**Table 1.** Number of heavy rain damage events and total heavy rain damage cost by city and province.

| Regional Division | Number of Heavy Rain Damage Events | Total Heavy Rain Damage Cost (Unit: 1 Million KRW) |
|---|---|---|
| **Gyeonggi-do** | **996** | **2,393,552** |
| Jeollanam-do | 762 | 608,826 |
| Gyeongsangbuk-do | 515 | 911,552 |
| Chungcheongnam-do | 440 | 467,558 |
| Gyeongsangnam-do | 429 | 1,259,151 |
| Gangwon-do | 421 | 3,405,380 |
| Jeollabuk-do | 421 | 636,105 |
| Seoul-si | 338 | 218,042 |
| Chungcheongbuk-do | 323 | 1,130,086 |
| Busan-si | 221 | 189,317 |
| Incheon-si | 211 | 79,273 |
| Daejeon-si | 98 | 51,143 |
| Gwangju-si | 90 | 42,507 |
| Ulsan-si | 75 | 44,366 |
| Jeju-do | 63 | 24,879 |
| Daegu-si | 37 | 8735 |
| Sejong-si | 32 | 11,995 |

The study selected Gyeonggi-do as the study area because the province reported the highest number of heavy rain damages and the highest damage cost. Figure 7 shows the current administrative districts of Gyeonggi-do. This province consists of 31 administrative districts, with diverse regional planning features that include urban, coastal, and rural areas. Therefore, the study assumes that this province would be a good representation of other places in the country.

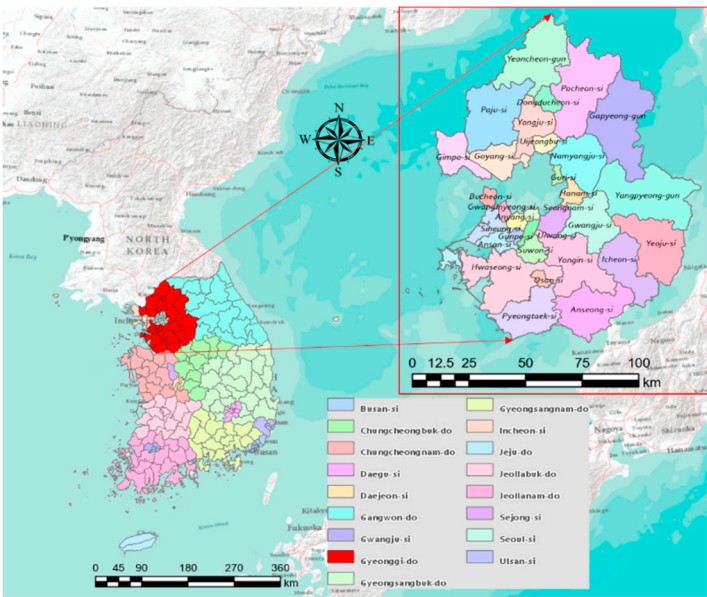

**Figure 7.** Study area (Gyeonggi-do).

*3.2. Dependent and Independent Variables*

3.2.1. Dependent Variables

To obtain the dependent variables that are required to develop the HDPM, the study collected data on heavy rain damage cost for each of the administrative districts. The data was gathered from the "Statistical Yearbook of Natural Disaster" which is annually published by MOIS after damage data compilation. The following are the pre-processing steps to construct the dependent variables from the

30 years (1988 to 2017) of damage data: (1) the Statistical Yearbook of Natural Disaster is based on the administrative districts of the corresponding year. Due to administrative changes like area integration and transfer, the researchers rezoned the area; (2) since the damage costs provided in the Statistical Yearbook of Natural Disaster reflect the monetary value of their corresponding year, it is required to convert them to its current value. For this reason, the study used the producer price index (PPI) and estimated the value conversion index by which the past value was multiplied to convert it to the 2017 value [39]; (3) the damage costs are dependent variables with ranges from 54,000 KRW (Korean Won) to 109,867,323,000 KRW. Due to the wide range of values, it will be difficult to develop an adequate model. Therefore, the study took the common logarithm ($\log_{10} x$) for the heavy rain damage costs and applied the algebraic conversion to reduce the data variability before developing the model. Table 2 shows the basic statistics of the dependent variables converted to the common logarithms. The range of values was narrowed to 1.73 as the minimum and 8.04 as the maximum with median and average values of 4.79 and 4.83, respectively.

**Table 2.** Basic statistics for dependent variables.

| Dependent Variable | | Min | Median | Mean | Max |
|---|---|---|---|---|---|
| Heavy rain damage | Total damage | 1.7324 | 4.7885 | 4.8285 | 8.0409 |

### 3.2.2. Independent Variables

The heavy rain damage prediction model (HDPM) based on linear regression requires rainfall data as its independent variables. To obtain such data, the study collected the hourly rainfall from the Automated Synoptic Observing System (ASOS) provided by the Weather Data Open Access Portal [40] of the Korea Meteorological Administration (KMA). The KMA operates the surface weather observation service to watch weather conditions by using the ASOS and the Automatic Weather System (AWS). There are 102 data points for ASOS and about 510 data points for AWS across the country. Thus, AWS data points are more densely distributed than those of ASOS. However, there is a limitation in using the AWS data because its equipment was only installed in 2000. This study used the hourly rainfall data from the ASOS for which weather observation has been continuously conducted for more than 30 years.

The spatial units for the data on heavy rain damage costs (which were constructed as dependent variables as explained in Section 3.2.1) are from the administrative districts while the collected rainfall data are from the weather observation stations. Therefore, it is required to integrate them based on the administrative districts. The researchers applied the Thiessen polygon method to convert the spatial units for the rainfall data. Figure 8 shows the Thiessen areal distribution applied for the study area. The Thiessen polygon method is used to calculate the area average rainfall for a study area using a weight, which is the area ratio of the Thiessen polygon formed by perpendicular bisectors of connecting lines between stations where the number of stations are located. Twelve ASOS stations have influenced the study area of Gyeonggi-do: Ganghwa station, Seoul station, Suwon station, Incheon station, Cheorwon station, Yangpyeong station, Icheon station, Cheonan station, Seosan station, Chuncheon station, Hongcheon station, and Wonju station. The Thiessen area over each of the administrative districts indicates the rainfall data station.

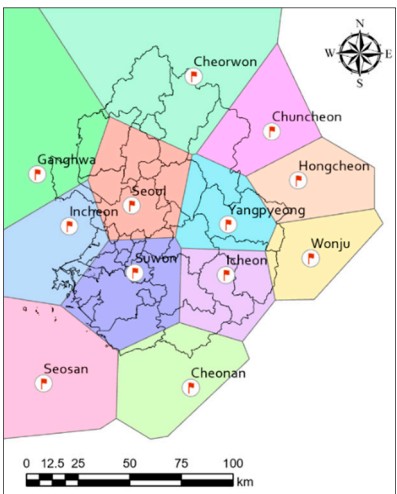

**Figure 8.** Thiessen area distribution of the study area.

The rainfall data are processed in three types: maximum rainfall (1 h to 24 h); total rainfall; and antecedent rainfall (−1 day to −7 days) as used by past studies [8,12,15,21,41,42]. The maximum rainfall by duration means the peak value of accumulated rainfall by duration (1 h to 24 h) that occurs during a disaster period. The total rainfall refers to the sum of rainfall for the entire period of the disaster period. The antecedent rainfall is a value obtained by accumulating the values of rainfall that occur from 1 to 7 days before the disaster. Table A1 shows the rainfall data (as independent variables) and their basic statistics. Since all of the data are related to rainfall, their units are in mm and the abbreviations are used for convenience in developing models.

*3.3. Development of HDPM*

3.3.1. HDPM Using Linear Regression Model

The use of data from the total dataset in developing a heavy rainfall damage model could produce a model that is suitable for the data but may not be applicable with future data (termed as "overfitting"). To prevent overfitting, the study classified the dataset into two groups: one for the training dataset which will develop the model and the test dataset which will evaluate the predictive performance of the developed model. The training dataset accounts for about 70% of the total dataset, and the other 30% for the test dataset. We used a linear regression model with damage costs as the dependent variables and rainfall data as the independent variables. A total of 32 rainfall data were used as independent variables in developing the HDPM. The linear regression models that can be produced are multitudinous, depending on the combination of the independent variables. Therefore, the process to select the independent variables are important to develop a linear regression model with high predictive performance.

The following three variable selection methods are widely used: forward selection method; backward elimination method; and stepwise selection method. The stepwise selection method supplements the shortcomings of the other two. By using this method, one can reconsider an explanatory variable that has been once selected or eliminated and select it in both ways. The stepwise method is highly stable because the variables that are selected or eliminated in the beginning are not fixed and can explore more cases compared to the other two methods. This study used the stepwise selection method, with a total of 12 independent variables selected: maximum rainfall by duration (2 h, 9 h, 13 h, 15 h, 16 h, and 20 h); total rainfall; and antecedent rainfall (1 day, 4 days, 5 days, 6 days, and 7 days). The HDPM derived from the linear regression model can be expressed as in Equation (7).

$$y = 3.5070 - 0.0066max2 - 0.0101max9 + 0.0291max13 + 0.0321max15$$
$$- 0.0662max16 + 0.0236max20 + 0.0015tot + 0.0075pre1 \tag{7}$$
$$+ 0.0051pre4 - 0.0122pre5 + 0.0146pre6 - 0.0053pre7$$

The high correlation between the independent variables made biased estimates of regression coefficients. Therefore, after the variables are selected, it is required to evaluate the multicollinearity of the variables that form the linear regression model. When the variance inflation factor (VIF) is higher than 10, it is assumed that the multicollinearity exists. As shown in Table A2, the VIFs were higher than 10 for the maximum rainfall (9 h, 13 h, 15 h, 16 h, and 20 h) and antecedent rainfall (−5 days, −6 days, and −7 days). So, for these variables, an issue on multicollinearity was noticed.

### 3.3.2. HDPM Using Principle Component Analysis and Regression Model

To relieve the multicollinearity problem of the HDPM developed from the linear regression model, we considered principle component analysis (PCA) that reduces the dimension of independent variables that have high correlation to an independent, small number of principal components. In order to properly include the information from the independent variables, we determined the number of principle components for the cumulative dispersion of more than 90%. As shown in Table A3, it was analyzed that the four principal components had more than 90% of cumulative dispersion. By using the four principle components obtained through the PCA, the study constructed the linear regression model and the stepwise selection method was applied in selecting the variables. The HDPM using PCA and the linear regression model can be expressed using Equation (8).

$$y = 4.8376 + 0.1495PC1 + 0.2291PC3 - 0.0598PC4 \tag{8}$$

The VIF calculations of the principal components yielded a value of 1, which indicates that there is no problem of multicollinearity.

## 4. Combined Heavy Rain Damage Prediction Model

### 4.1. Dependent and Independent Variables

#### 4.1.1. Dependent Variables for RPM

For the HDPM, the relationship between heavy rain damage (dependent variables) and rainfall data (independent variables) is simply assumed to be linear. Extensively, there is a limitation that it cannot explain nonlinear relations. In addition, it is not able to reflect social and economic characteristics of a region that have an indirect influence on heavy rain damages. For this reason, there is a tendency in which a significant difference (residual) occurs between an actual and the predicted damage cost. In Section 2.3, we proposed the residual prediction model (RPM) that can predict the residual from the HDPM. More specifically, it constructs the combined heavy rain damage prediction model (CHDPM) for better predictive performance. CHDPM proposes a predicted heavy rain damage cost as a combination of the predicted damage cost provided by the HDPM and the predicted residual by the RPM. Therefore, the residual from the HDPM was used as the dependent variable for the RPM. The residuals range from −3.4 to 3.4 and the mean (−0.02) and median (−0.03) values were close to zero (0).

#### 4.1.2. Independent Variables for RPM

By referring to past related studies, the study considered 10 social and economic data that are known to have an indirect influence on heavy rain damage as independent variables of the RPM [15,21,22,30]. The 10 data include gross regional domestic product, financial independence rate, population density, population, area, dilapidated dwelling rate, number of houses, number of dilapidated dwelling, processing capacity of pumps, and number of pumps. The gross regional domestic product means an added value of a study area that is calculated with basic statistics including

production, consumption, and price. The higher the gross regional domestic product, the higher income of the area. In this regard, the area is regarded as well prepared for heavy rain damages, however, because its asset value is relatively high, there is a risk that the damage cost will increase. Financial independence rate refers to a share of financial resources (revenues) that a local government secures independently for its general accounting revenue. It is an indicator that represents what level of the local government raises funds required for financial activities. Therefore, the financial independence rate is used to explain the capacity for independent financial operation of a local government. The higher rate indicates solid financial independence and thus was assumed to reduce damages from heavy rain. Population density explains how dense the study area is by the ratio of population to the area. It was assumed that the higher population in a certain area will cause more damages even if the area is small. Dilapidated dwelling rate is the ratio of dwelling aged over 30 years vulnerable to heavy rain to the number of dilapidated dwellings. The study considered that the higher dilapidated dwelling rate indicates the target region is deprived (e.g., old downtown) and at higher risk for large scale damage. The processing capacity of pumps is obtained by accumulating processing capacities of discharge pumps installed in the area. We assumed that the higher capacity and number of pumps would mitigate the damages.

Table A4 shows the list and the basic statistics of social and economic characteristic data that were used as independent variables in the study. Since the variables' units and range of values are different, a re-scaling method for standardization is used to have values from 0 to 1. Re-scaling uses the maximum and minimum values of the variables for standardization and can be explained as shown in Equation (9).

$$\text{Re} - \text{scaling} \ = \ \frac{X_i - \min(X)}{\max(X) - \min(X)} \tag{9}$$

*4.2. Development of RPM*

4.2.1. RPM Using Decision Tree

In each step of pruning, we checked the values of CP(complexity parameter) statistics based on 10-fold cross validation. CP statistics is the estimate of a standardized residual sum of squares used to model selection. A lower CP value ensures better predictive performance. The depth at which the lowest CP value was determined was the optimal value and we finally applied this depth to the pruning. Figure 9a is the decision tree before pruning and Figure 9b is the final decision tree after the pruning was made to prevent overfitting. The text above each node indicates how the split of the tree was performed. The number above in each node is the average of the response variables included in that node. The lower left *n* means the number of data included in the node, and the right side % means the ratio. The former predicted the residual with population density, dilapidated dwelling rate, number of dilapidated dwellings, population, and financial independence rate. However, the latter used population density only for residual prediction. For the decision tree, the population density was considered as the most important factor for the prediction.

4.2.2. RPM Using Random Forest

In Figure 10a, the horizontal axis represents the bootstrap samples (size of tree) and the vertical axis represents the OOB error. The number of bootstrap samples was 209, which minimizes the OOB error of Figure 10a. Figure 10b shows the importance for each of the independent variables. According to the analysis, the most important variables for predicting the residual are population density, area, financial independence rate, gross regional domestic product, and population.

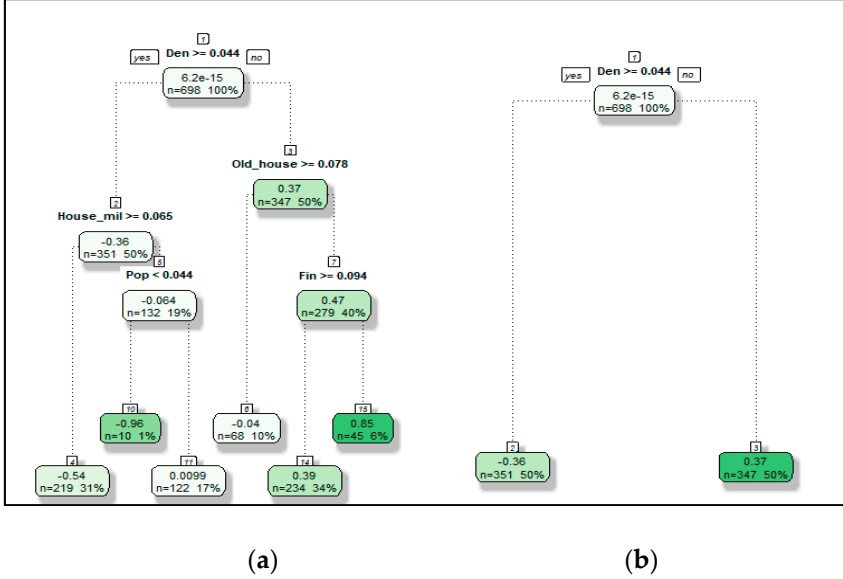

**Figure 9.** (**a**) Residual prediction model (RPM) using decision tree (before pruning); (**b**) RPM using decision tree (after pruning).

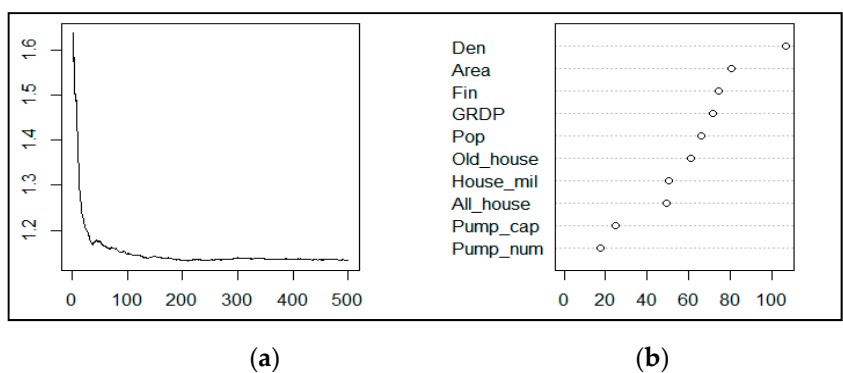

**Figure 10.** (**a**) Error by size of tree; (**b**) importance by variables.

### 4.2.3. RPM Using Support Vector Regression

This study considered radial basis function (RBF) as a kernel function for training support vector regression. Polynomial and sigmoid functions were also considered as a kernel. However, they showed worse predictive performance than RBF; we only presented the result of the RBF. The SVM has several hyper-parameters and we selected the optimal values minimizing $\epsilon$-insensitive loss function by cross validation [43]. Referring to existing studies, this study used the 10-fold cross validation and considered some candidate parameters in the range of costs in $2^k$, $k = 0, 1, \ldots, 7$ and the range of $\epsilon$ from 0 to 1 by intervals of 0.1 for the optimal choice [44]. As a result, the error is minimal when the cost is 1 and the $\epsilon$ value is 0.9.

### 4.2.4. RPM Using Deep Neural Networks

In this study, the deep neural networks (DNN) applied the resilient backpropagation with backtracking instead of the backpropagation as it is known to be a more solid version of the algorithm [45]. Also, the logistic activation function was used. The error function was used with the sum of the square errors and the threshold was set to 0.1. The DNNs can determine the number of hidden layers and the number of nodes. In general, with the increasing number of hidden layers, the neural network gets deeper and more complex so the abstract features can be extracted. However, a higher complexity may lead to the tendency of the local minimum or significant increase in calculation

processes. Therefore, the study sets the number of hidden layers as 3 and for each of the layers, 1 to 5 nodes were prepared to construct a total of 125 ($5 \times 5 \times 5 = 125$) DNNs. In the 125 constructed DNNs, the smallest RMSE was found in the DNN composition with four nodes for the first hidden layer, three nodes for the second hidden layer, and three nodes for the third hidden layer. Therefore, we used the deep neural networks to compose the RPM as shown in Figure 11. The standardized prediction was then re-scaled to convert it into the state before the standardization process.

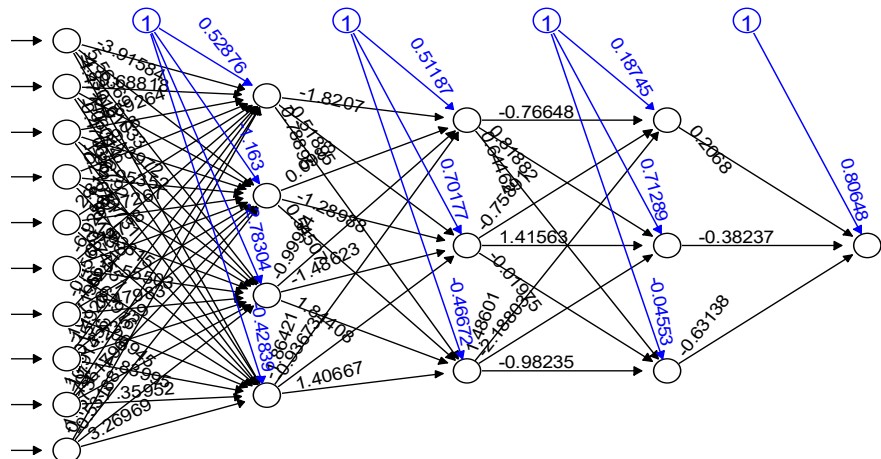

**Figure 11.** RPM using deep neural networks (black: weights, blue: biases).

### 4.3. Evaluation of Predictive Performance by Models

To evaluate the predictive performance of the HDPM and the CHDPMs that are modeled with the RPM, this study used a test dataset that accounts for 30% of the total dataset and has not been used for model development. To evaluate the trained CHDPMs, this study calculated predicted residuals of the RPMs combined with the predicted heavy rain damage of the HDPM provided in Section 3 to produce the final heavy rain damage prediction. The data of the test dataset were applied to the HDPM and CHDPMs for comparison between the actual heavy rain data cost of the test dataset and the derived prediction. The RMSE was 1.04 for the HDPM only based on the linear regression model, while it was 0.96 for the CHDPM with the decision tree, 1.01 for the CHDPM with random forest, and 0.94 for the CHDPM with the SVM, indicating the predicted error decreased from the level of the conventional HDPM. In a similar vein, the sMAPE that shows the percentage of error in the predicted value was 19% for the HDPM, while it was 17% for the CHDPM (DT) and CHDPM (RF) and 16% for the CHDPM (DT). The cor. that helps identify the tendency in actual and predicted values was 63% for HDPM, 70% for CHDPM (DT) and the CHDPM (DNN), and 71% for the CHDPM (SVM) (in Bold), indicating that the CHDPMs catch the tendency of actual values better than the HDPM. To sum up the results of the three indicators for predictive performance evaluation, the CHDPM (SVM) that combined the RPM with the SVM showed the most excellent predictive performance. Table 3 summarizes the comparison of predictive performances with the indicators.

**Table 3.** Evaluation of predictive performance by models.

| Index | HDPM | CHDPM(DT) | CHDPM(RF) | CHDPM(SVM) | CHDPM(DNN) |
|-------|------|-----------|-----------|------------|------------|
| RMSE | 1.0429 | 0.9567 | 1.0051 | **0.9398** | 1.2067 |
| sMAPE | 0.1871 | 0.1651 | 0.1705 | **0.1626** | 0.2038 |
| cor. | 0.6293 | 0.7017 | 0.6829 | **0.7145** | 0.7059 |

## 5. Discussion

### 5.1. Summary

The study developed the combined heavy rain damage prediction model (CHDPM), which is supplementary to the heavy rain damage prediction model (HDPM), with the residual prediction model (RPM) using the machine learning techniques (DT, RF, SVM, and DNN), in order to increase the predictive performance of the HDPM. The RMSE of the HDPM using the linear regression model was analyzed as 1.04, sMAPE was 19%, and cor. was 63%. With respect to the RDM (DT) developed using the DT, population density was found to be the most important factor to predict residuals. The CHDPM (DT), a model with the addition of the RPM (DT), showed an increase in performance by 8% for the RMSE, 12% for the sMAPE, and 12% for the cor., from the level of the HDPM. For the RPM (RF), population density was identified as the most important factor to predict residuals, followed by area and financial independence. The analysis result of the CHDPM (SVM), a model added with RPM (SVM), indicated an increase in performance by 4% for the RMSE, 9% for the sMAPE, and 9% for the cor., from the level of the HDPM. With respect to the RPM (SVM) developed using the SVM, the model had a limitation in identifying the importance of the variables. However, the predictive performance of the CHPDM (SVM), a model with the addition of the RPM (SVM), was increased by 10% for the RMSE, 13% for the sMAPE, and 14% for the cor., which had the highest level of improvement. The CHDPM (DNN), a model combined with the RPM (DNN) using the DNN, had no increase in the performance for the RMSE and sMAPE, however there was an improvement by 12% for the cor. As a result of analyzing the three indicators for the predictive performance evaluation, the CHDPM using the SVM was identified as the combined model with the highest predictive performance. Therefore, this study selected the CHDPM (SVM) as the final model. Table 4 shows improvements in predictive performance of the CHDPMs, in comparison with the HDPM for each of the indicators.

**Table 4.** Predictive performance improvement rate (%).

| Index | CHDPM (DT) | CHDPM (RF) | CHDPM (SVM) | CHDPM (DNN) |
|-------|------------|------------|-------------|-------------|
| RMSE | +8.2656 (%) | +3.6211 (%) | **+9.8881 (%)** | −15.7050 (%) |
| sMAPE | +11.7575 (%) | +8.8692 (%) | **+13.1333 (%)** | −8.8805 (%) |
| cor. | +11.5103 (%) | +8.5180 (%) | **+13.5353 (%)** | +12.1712 (%) |

### 5.2. Any Other Model

In this study, only the rainfall data were used as the independent variables for the HDPM. In the case of CHDPMs, social and economic data were additionally used for the development of the RPMs. One may suggest that it is not the combination with the RPMs that predict residuals but the additional consideration for social data to the independent variables which lead to the improvement of models' predictive performance. Responding to this, the study defined another new heavy rain damage prediction function as HDPM2 (heavy rain damage prediction model ver.2), using the machine learning model for which the rainfall data is the independent variable for the HDPM and the social and economic data are the independent variables for the RPM. The same predictive performance evaluation was conducted for the HDPM2. Table 5 provided the result of comparison between the CHDPM (SVM) and the HDPM2. It was found that in general, the CHDPM (SVM), the HDPM with the addition of the residual prediction model that uses the SVM, had a better predictive performance than the prediction model for which the individual variables were additionally considered. This indicates the applicability of applying machine learning, which accurately predicts residuals, to the conventional linear regression model rather than simply adding independent variables.

**Table 5.** Evaluation of predictive performance with any other model.

| Index | CHDPM (SVM) | HDPM2 (DT) | HDPM2 (RF) | HDPM2 (SVM) | HDPM2 (DNN) |
|---|---|---|---|---|---|
| RMSE | **0.9398** | 1.0785 | 0.9677 | 1.0347 | 1.0260 |
| sMAPE | **0.1626** | 0.1833 | 0.1615 | 0.1699 | 0.1670 |
| cor. | **0.7145** | 0.5927 | 0.6744 | 0.6189 | 0.6372 |

*5.3. Future Research Directions*

The study predicts heavy rain damages using machine learning models to predict residuals that are not explicit with conventional linear regression models alone. These residuals were combined with the model for the purpose of enhancing its predictive performance. Table 6 provides the result of comparing the conventional HDPM and the CHDPM (SVM) developed by this study, with the heavy rain damage cost (log) as the dependent variable, being calculated back for the conversion into units (1000 KRW) of actual costs. The CHDPMs are expected to propose more accurate forecasting than the HDPM, however, they showed a limitation in considering the damage reduction effect from the disaster prevention projects that are implemented by the region such as levee maintenance and construction of flood forecasting system and the factors that aggravate the damages that include deteriorated facilities and absence of people in charge. Therefore, the predictive performance would be improved further if those effects or aggravating factors were reflected in the models. In addition, the study used the limited number of rainfall data using weather data, however, the use of other weather data associated with heavy rain damage such as temperature, humidity, cloud cover or radar rainfall estimations can be considered to enhance the predictive performance. Since it was our first trial to develop a combination type of model for heavy rain damage prediction, we simply combined the linear regression model that uses rainfall data with the machine learning models that have independent variables like social and economic factors. Future studies can consider other ways to combine the two different machine learning models.

**Table 6.** Evaluation of predictive performance by models (real scale).

| | RMSE | sMAPE | Cor. |
|---|---|---|---|
| HDPM | 9,285,423 | 1.2836 | 0.1559 |
| CHDPM(SVM) | 8,827,218 | 1.1452 | 0.3619 |

**Author Contributions:** Conceptualization, C.C. and J.K. (Jeonghwan Kim); data curation, C.C. and J.K. (Jungwook Kim); formal analysis, C.C.; funding acquisition, H.S.K.; investigation, C.C.; methodology, J.K. (Jeonghwan Kim); project administration, H.S.K.; resources, C.C. and J.K. (Jungwook Kim); software, C.C.; supervision, J.K. (Jeonghwan Kim) and H.S.K.; validation, J.K. (Jeonghwan Kim); visualization, C.C. and J.K. (Jeonghwan Kim); writing—original draft, C.C., and J.K. (Jeonghwan Kim); writing—review and editing, J.K. (Jeonghwan Kim) and H.S.K.

**Funding:** This research was supported by a grant (MOIS-DP-2015-05) from the Disaster Prediction and Mitigation Technology Development Program funded by the Ministry of the Interior and Safety (MOIS, Korea).

**Conflicts of Interest:** The authors declare no conflicts of interest.

# Appendix A

**Table A1.** Basic statistics of independent variables.

| Independent Variables | | Min (mm) | Median (mm) | Mean (mm) | Max (mm) |
|---|---|---|---|---|---|
| Maximum rainfall by duration (1 h) | max1 | 0 | 27.92 | 30.14 | 96.00 |
| Maximum rainfall by duration (2 h) | max2 | 0 | 44.42 | 47.94 | 158.12 |
| Maximum rainfall by duration (3 h) | max3 | 0 | 56.06 | 60.87 | 208.50 |
| Maximum rainfall by duration (4 h) | max4 | 0 | 64.00 | 70.93 | 254.85 |
| Maximum rainfall by duration (5 h) | max5 | 0 | 70.83 | 78.72 | 316.17 |
| Maximum rainfall by duration (6 h) | max6 | 0 | 77.52 | 85.55 | 341.31 |
| Maximum rainfall by duration (7 h) | max7 | 0 | 81.96 | 91.53 | 354.17 |
| Maximum rainfall by duration (8 h) | max8 | 0 | 86.93 | 97.21 | 371.36 |
| Maximum rainfall by duration (9 h) | max9 | 0 | 91.05 | 102.59 | 383.89 |
| Maximum rainfall by duration (10 h) | max10 | 0 | 94.75 | 106.81 | 401.08 |
| Maximum rainfall by duration (11 h) | max11 | 0 | 98.50 | 110.43 | 420.51 |
| Maximum rainfall by duration (12 h) | max12 | 0 | 102.00 | 113.70 | 445.60 |
| Maximum rainfall by duration (13 h) | max13 | 0 | 103.99 | 116.63 | 457.60 |
| Maximum rainfall by duration (14 h) | max14 | 0 | 105.66 | 119.58 | 475.36 |
| Maximum rainfall by duration (15 h) | max15 | 0 | 107.55 | 122.39 | 486.03 |
| Maximum rainfall by duration (16 h) | max16 | 0 | 109.69 | 125.09 | 490.97 |
| Maximum rainfall by duration (17 h) | max17 | 0 | 111.83 | 127.32 | 492.08 |
| Maximum rainfall by duration (18 h) | max18 | 0 | 114.00 | 129.59 | 492.94 |
| Maximum rainfall by duration (19 h) | max19 | 0 | 115.90 | 131.79 | 493.38 |
| Maximum rainfall by duration (20 h) | max20 | 0 | 116.91 | 134.07 | 493.54 |
| Maximum rainfall by duration (21 h) | max21 | 0 | 118.90 | 136.36 | 494.77 |
| Maximum rainfall by duration (22 h) | max22 | 0 | 120.10 | 138.18 | 495.63 |
| Maximum rainfall by duration (23 h) | max23 | 0 | 122.04 | 139.98 | 496.00 |
| Maximum rainfall by duration (24 h) | max24 | 0 | 123.86 | 141.79 | 496.16 |
| Total rainfall | tot | 0 | 194.40 | 243.80 | 1202.40 |
| Antecedent rainfall (1 day before) | pre1 | 0 | 0.38 | 5.97 | 82.75 |
| Antecedent rainfall (2 days before) | pre2 | 0 | 3.72 | 14.43 | 190.00 |
| Antecedent rainfall (3 days before) | pre3 | 0 | 10.76 | 25.52 | 203.04 |
| Antecedent rainfall (4 days before) | pre4 | 0 | 17.26 | 36.13 | 242.19 |
| Antecedent rainfall (5 days before) | pre5 | 0 | 30.27 | 49.43 | 418.00 |
| Antecedent rainfall (6 days before) | pre6 | 0 | 38.04 | 57.70 | 419.50 |
| Antecedent rainfall (7 days before) | pre7 | 0 | 45.89 | 65.85 | 419.50 |

**Table A2.** Variance inflation factor by independent variables.

| Independent Variable | VIF | Independent Variable | VIF |
|---|---|---|---|
| Maximum rainfall by duration (2 h) | 4.8146 | Total rainfall | 2.5020 |
| **Maximum rainfall by duration (9 h)** | **69.7583** | **Antecedent rainfall (1 day ago)** | **1.0921** |
| **Maximum rainfall by duration (13 h)** | **339.4480** | **Antecedent rainfall (4 days ago)** | **3.8679** |
| **Maximum rainfall by duration (15 h)** | **966.7931** | **Antecedent rainfall (5 days ago)** | **14.3081** |
| **Maximum rainfall by duration (16 h)** | **868.9390** | **Antecedent rainfall (6 days ago)** | **27.3899** |
| **Maximum rainfall by duration (20 h)** | **90.2535** | **Antecedent rainfall (7 days ago)** | **14.8949** |

**Table A3.** Proportion of variance by principle components.

| Principle Components | Standard Deviation | Proportion of Variance | Cumulative Proportion |
|:---:|:---:|:---:|:---:|
| **PC1** | **4.7696** | **0.7109** | **0.7109** |
| **PC2** | **2.1376** | **0.1428** | **0.8537** |
| **PC3** | **1.1823** | **0.0437** | **0.8974** |
| **PC4** | **1.0768** | **0.0362** | **0.9336** |
| PC5 | 0.8601 | 0.0231 | 0.9567 |
| PC6 | 0.6997 | 0.0153 | 0.9720 |
| PC7 | 0.5236 | 0.0086 | 0.9806 |
| ⋮ | ⋮ | ⋮ | ⋮ |

**Table A4.** Summary statistics of independent variables.

| Independent Variables | | Min (mm) | Median (mm) | Mean (mm) | Max (mm) |
|:---:|:---:|:---:|:---:|:---:|:---:|
| Gross regional domestic product | GRDP | 0 | 0.0792 | 0.1487 | 1 |
| Financial independence rate | Fin | 0 | 0.4055 | 0.4089 | 1 |
| Population density | Den | 0 | 0.0445 | 0.1446 | 1 |
| Population | Pop | 0 | 0.1345 | 0.2442 | 1 |
| Area | Area | 0 | 0.4412 | 0.4041 | 1 |
| Dilapidated dwelling rate | House_mil | 0 | 0.1594 | 0.1919 | 1 |
| Number of houses | All_house | 0 | 0.1536 | 0.2473 | 1 |
| Number of dilapidated dwelling | Old_house | 0 | 0.0424 | 0.0881 | 1 |
| Processing capacity of pumps | Pump_cap | 0 | 0.0056 | 0.0809 | 1 |
| Number of pumps | Pump_num | 0 | 0.0526 | 0.1393 | 1 |

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
