# Peer review of "Development of Combined Heavy Rain Damage Prediction Models with Machine Learning"

_water, doi:10.3390/w11122516_

Round 1

Reviewer 1 Report

This article is well written and really interesting to read even for a non specialist as me. I think that section 2.2 could be shrinked a little in order to focus more on the results. The database used for the validation of the methods appear as complete and well dscribed in section 3.1. One could regret the limited number of plviometers in the area and the authors could discuss the possibility to use radar rainfall estimations. Finally figures 10 and 11 are not easily readable and must be improved. I recommand publication of this contribution.

Author Response

1. I think that section 2.2 could be shrinked a little in order to focus more on the results.
=> Section 2.2 has been rewritten to make it easier and shorter to understand.

2. One could regret the limited number of pluviometers in the area and the authors could discuss the possibility to use radar rainfall estimations.
=> In Section 5.3, we have added the contents that can use radar rainfall data.

3. Finally figures 10 and 11 are not easily readable and must be improved.
=> To make it easier to understand Figure 10, we added contents and improved the resolution.

Reviewer 2 Report

Dear Author

You did a great job and highlight the important topic. This is a very good addition in this field however I have a few concerns as follow:

Comments:

Introduction:

Line 25: Rephrase the introduction line. Something like the intensity and frequency of extreme events has been increased worldwide due to global warming.

Line 28-29: Prediction of the extent of natural disaster damages based on historical data will greatly help in the formulation of countermeasures to reduce the damages (How)? Prediction is based on future datasets not on historical, from historical data you can track the trend of disasters.

Line 57: What socioeconomic data were selected and used for this study in your modelling as an independent variable please mention them at least.

Figure 1: is very good descriptive but not clear. Please make it good in terms of its resolution as it is difficult to read properly if possible use light colours (suggestion)

Line 72: Can you please explain a bit of VIF. Also, no idea where the author selects the value of 10.

Line 93: about the predictive value the process is complicated please write it down in simple words or explain the process?

Line 95-98: Very complex and confusing. As a reviewer, I am lost in the mid of the ocean. Please write it in simple words so that someone else can also follow the process.

Line 180-220: Model is not well explained and methodology is bit complex and confusing. It is therefore suggested to use simple language and explain a bit.

Figure-7 is difficult to read; please make it clear and if possible divide it into sections that explain it.

Line263-280 again the Dependent variable section is very complex very difficult to understand.

Line 283: Table-2 there is no need to put this table it can be explained in text form as well. Remove the table.

Figure-8: very difficult to read must improve the resolution.

Table-4, 5, 6 and 9: should be supplementary material.

Table 7 and 8 should be removed and explain in words.

Figure 9 is again not clear. Please improve the resolution and make it visible.

Line 493: Summary should be written in paragraphs rather than in bullet points.

Author Response

* Thank you for good comment. Your comments helped us improve the quality of the paper.

Line 25: Rephrase the introduction line. Something like the intensity and frequency of extreme events has been increased worldwide due to global warming.
Line 28-29: Prediction of the extent of natural disaster damages based on historical data will greatly help in the formulation of countermeasures to reduce the damages (How)? Prediction is based on future datasets not on historical, from historical data you can track the trend of disasters.
Line 57: What socioeconomic data were selected and used for this study in your modelling as an independent variable please mention them at least.
Figure 1: is very good descriptive but not clear. Please make it good in terms of its resolution as it is difficult to read properly if possible use light colours (suggestion)
=> As you said, the Introduction has been revised and marked in yellow.

Line 72: Can you please explain a bit of VIF. Also, no idea where the author selects the value of 10.
=> Additional details were written about the VIF and marked in yellow.

Line 93: about the predictive value the process is complicated please write it down in simple words or explain the process?
Line 95-98: Very complex and confusing. As a reviewer, I am lost in the mid of the ocean. Please write it in simple words so that someone else can also follow the process.
Line 180-220: Model is not well explained and methodology is bit complex and confusing. It is therefore suggested to use simple language and explain a bit.
=> As you said, we have corrected the whole text to the correct grammar and scientific expression. And marked in yellow.

Figure-7 is difficult to read; please make it clear and if possible divide it into sections that explain it.
=> As you said, we have corrected the whole text to the correct grammar and scientific expression. And marked in yellow.

Line263-280 again the Dependent variable section is very complex very difficult to understand.
=> Modifications were made to make it easier to understand about Dependent variable section. And marked in yellow.

Line 283: Table-2 there is no need to put this table it can be explained in text form as well. Remove the table.
Table-4, 5, 6 and 9: should be supplementary material.
Table 7 and 8 should be removed and explain in words.
=> I have deleted Table 2, 7 and 8 according to your advice. Also table 4, 5, 6 and 9 moved Appendix A. And marked in yellow.

Figure-8: very difficult to read must improve the resolution.
Figure 9 is again not clear. Please improve the resolution and make it visible.
=> We have improved the resolution in Figure 8 and Figure 9. Also added some content to make it easier to understand. And marked in yellow.

Line 493: Summary should be written in paragraphs rather than in bullet points
=> As you said, we have corrected the whole paragraphs. And marked in yellow.